# Calreticulin—From the Endoplasmic Reticulum to the Plasma Membrane—Adventures of a Wandering Protein

**DOI:** 10.3390/cancers17020288

**Published:** 2025-01-17

**Authors:** Gillian C. Okura, Alamelu G. Bharadwaj, David M. Waisman

**Affiliations:** 1Department of Pathology, Dalhousie University, Halifax, NS B3H 1X5, Canada; gillian.okura@dal.ca (G.C.O.); alamelu.bharadwaj@dal.ca (A.G.B.); 2Department of Biochemistry and Molecular Biology, Dalhousie University, Halifax, NS B3H 1X5, Canada

**Keywords:** calreticulin, myeloproliferative neoplasms, plasminogen, phagocytosis, high-affinity calcium-binding protein

## Abstract

This review examines the chronological timeline of studies identifying and characterizing calreticulin (CRT) from its origin as an endoplasmic reticulum (ER) protein to an important cell surface signaling molecule. A detailed literature analysis shows that CRT was initially discovered as the Ca^2+^-binding ER protein, called calregulin, and not as the SR protein(s), called the high-affinity calcium-binding protein (HACBP). We further elucidate the critical functions of CRT in calcium homeostasis, protein folding, immunogenic cell death (ICD), and antigen presentation. The roles of CRT in the regulation of the destruction of cancer cells by the immune system and as a causative factor in certain blood cancers are also discussed. Overall, this review critically and comprehensively identifies the original studies that revealed CRT’s discovery, structure, and functions in the ER and at the cell surface.

## 1. Introduction

Calreticulin (CRT), a highly conserved 46 kDa protein, has received considerable attention as a multifunctional molecule with diverse roles in intracellular and extracellular processes. Initially identified as a calcium-binding protein in the sarcoplasmic reticulum (SR), CRT’s significance has expanded far beyond its initial characterization. This protein is now understood to be an endoplasmic reticulum (ER)-resident protein, where it plays critical roles in calcium homeostasis and protein folding. Structurally, CRT consists of three distinct domains: the N-terminal domain, the proline-rich P-domain, and the C-terminal domain, each contributing to its adaptable functionality. Beyond its intracellular roles, CRT has been found to operate in extracellular spaces, on cell surfaces, and even in the cytosol and nucleus, participating in processes such as calcium homeostasis, chaperone function and protein folding, antigen presentation, and immune response regulation, especially during immunogenic cell death (ICD). CRT has also been shown to participate in the recognition of cancer cells by immune cells and in the phagocytosis of cancer cells. Mutations in CRT have been identified in myeloproliferative neoplasms (MPNs), which have contributed to our understanding of the etiology of this disease.

This review presents a new perspective on the historical events that led to the discovery of CRT. We also delineate the chronological events that led to the discovery of CRT and highlight the key original studies that have led to our knowledge of its structure and function. We also discuss the structural features and multifaceted functions of CRT in both intracellular and extracellular contexts and focus on the established functions of CRT in cancer.

## 2. Calreticulin-Enigmatic Discovery

It has been more than 40 years since the report of the HACBP as a Ca^2+^-binding protein of the rabbit skeletal muscle SR. This section introduces the concept that although countless publications quote the HACBP as the first report of CRT, a rigorous analysis of the publications that followed that report suggests that the HACBP was likely a mixture of several proteins. Furthermore, the biochemical characterization of the HACBP presented in that publication indicates that it is distinct from CRT.

### 2.1. Identification of the HACBP in Skeletal Muscle SR

It has been suggested that calreticulin (CRT) was originally identified as a Ca^2+^-binding protein of the rabbit muscle SR, called the high-affinity Ca^2+^-binding protein (HACBP) [1]. We have conducted a rigorous literature analysis of papers that have characterized the Ca^2+^-binding proteins of the rabbit skeletal muscle SR [2]. Because none of these laboratories has detected the HACBP or CRT in skeletal muscle SR preparations (Table 1), we have concluded that CRT is not the HACBP and is also not present in the rabbit skeletal muscle SR [2].

We were also unable to find supportive evidence for the existence of the HACBP as a single protein, as it has been shown convincingly that the band in the SDS-PAGE analysis of the rabbit skeletal muscle SR, reported by MacLennan and called the HACBP, was actually composed of three to five proteins [3,4,5]. Furthermore, the characterization of the HACBP by the MacLennan laboratory [5,6,7] differs significantly from the known biochemical properties reported for CRT, suggesting that the HACBP and CRT are distinct proteins. Our comprehensive analysis of the published studies on the SR’s Ca^2+^-binding proteins from other laboratories has forced the rewriting of the dogma that CRT is present in the rabbit muscle SR and was initially identified as the HACBP [2]. In this section, we summarize our findings and critically evaluate the events leading to the discovery of the HACBP and the suggestion that the HACBP is CRT.

In 1970, the MacLennan laboratory published a procedure for purifying the rabbit muscle SR [8]. When analyzed using the Weber and Osborn SDS-PAGE procedure [9], their SR preparation resolved into seven major Coomassie blue staining bands. The MacLennan laboratory identified the proteins corresponding to four of these bands as the Ca^2+^ ATPase (100 kDa), calsequestrin (44 kDa), a 30 kDa protein, and a proteolipid (12 kDa). The MacLennan laboratory also identified a prominent 55 kDa Coomassie blue staining band, which they assumed contained a single protein. The 55 kDa band was called the HACBP and characterized in 1974. To purify the HACBP, they subjected the SR preparation to four column-chromatography steps. SDS-PAGE was used to analyze the chromatography elution profiles, and the fractions containing the 55 kDa Coomassie blue staining band were pooled. The pooled 55 kDa fractions were named the HACBP and characterized as a Ca^2+^-binding protein [7]. That year, MacLennan also reported that the HACBP comprised 5–10% of the SR proteins, while calsequestrin comprised about 15–20% [6].

Then, in 1979, the MacLennan laboratory reexamined the contribution of the HACBP to the 55 kDa band. For this analysis, they used the Laemmli slab gel system and a two-dimensional slab gel system utilizing a Weber–Osborn separation in one dimension and a Laemmli separation in the second dimension [10]. They found four protein bands in the 55 kDa region, the most prominent of which was a diffusely stained protein of about 53 kDa [10]. However, which of these four bands corresponded to the HACBP was not reported. Two postdoctoral fellows at the MacLennan Laboratory, Dr. Marek Michalak and Dr. Kevin Campbell, also reanalyzed the rabbit muscle SR proteins using the Laemmli SDS-PAGE method [11]. They confirmed that the Weber and Osborn SDS-PAGE procedure used for detecting and purifying the HACBP had failed to resolve several proteins of similar molecular weights present in the HACBP band. They utilized a high-resolution strategy to separate the SR proteins: using Weber and Osborn SDS-PAGE in the first dimension and Laemmli SDS-PAGE in the second dimension. This procedure resolved the 55 kDa staining band into multiple proteins. Campbell reported three of these proteins: HCABP-1 (53 kDa), HACBP-2 (55 kDa), and HACBP-3 (56 kDa) [12] (reviewed by Campbell in [5]). The most prominent of these proteins, HACBP-1, comprised more than 90% of the 55 kDa band and was called the 53 kDa glycoprotein [4]. The other proteins, HACBP-2 and HACBP-3, were not purified or characterized, so it was unclear which of these bands corresponded to the HACBP. Michalak reported the presence of at least five proteins present in the 55 kDa band. He also noted that pyruvate kinase comprised about 2% of that band and that the mitochondrial F1 subunits were also present [3]. Therefore, Michalak’s data suggest that the HACBP should comprise about 5% of the 55 kDa proteins. Although Michalak alluded to the HACBP as being one of the five 55 kDa proteins of the rabbit skeletal muscle SR, the HACBP was never purified from the other 55 kDa proteins and characterized.

The possibility that the 55 kDa fraction purified and characterized by MacLennan [6,7,13,14] contained multiple proteins was likely because MacLennan used the Weber and Osborn SDS-PAGE method to detect and pool this fraction, and this method could not discriminate between any of the five proteins in the HACBP band. The basic difference between these two methods is that the Weber and Osborne method is a continuous electrophoresis method in which proteins are electrophoresed in a single buffer. In contrast, the Laemmli system is a discontinuous system that utilizes a stacking gel on top of a resolving gel. The stacking gel generates a tight voltage gradient between the leading edge of one buffer and the trailing edge of a second buffer, resulting in proteins of different charges forming narrow bands or discs, driven by the voltage gradient at the discontinuity. As such, the Laemmli procedure results in tighter protein bands.

MacLennan reported the purification of about 6 mg of calsequestrin from 800 mg of rabbit muscle, using three chromatography columns [15]. They utilized four chromatography columns to purify the HACBP and reported the yield as 1 mg/800 g skeletal muscle tissue [7]. This suggests a relative ratio of about 1:5 for the HACBP/calsequestrin. The Weber and Osborn SDS-PAGE analysis presented by MacLennan [7] showed that the relative staining intensity between the band containing calsequestrin and the HACBP band was about 5/1. If the 1 mg of HACBP was pure HACBP/CRT and considering that it comprises 5% of the 55 kDa band, one would expect that the SR preparation would contain (100%/5% × 1 mg) 20 mg of 55 kDa HACBP-band proteins. Thus, the 55 kDa HACBP band would be expected to be (20 mg/6 mg), about three times the staining intensity of calsequestrin, which is not observed (Figure 1A).

In his extensive historical review of the rabbit skeletal muscle SR proteins, Campbell addressed the issue of the content of the 55 kDa Coomassie blue staining band that was initially reported by MacLennan and named the HACBP. The SDS-PAGE analysis of the MacLennan rabbit skeletal muscle SR preparation, reanalyzed by Campbell, is presented in Figure 1. As indicated, the 53 kDa glycoprotein was the dominant protein in the 55 kDa region. Notably, Stains-all did not stain any proteins in the 55 kDa region (Figure 1B). Similarly, Van et al. [16] reported that although the intense Stains-all staining of calsequestrin was observed in their SR preparation, no Stains-all staining was detected in the 55 kDa region of the gel, i.e., CRT was not detected. These authors, however, observed the Stains-all staining of the CRT in other tissues (Figure 2).

It is interesting that Campbell reported that the purified HACBP(s) did not bind Stains-all [5] (Figure 1B). Campbell pioneered Stains-all for the staining of Ca^2+^-binding proteins and demonstrated that Stains-all was a more sensitive stain for Ca^2+^-binding proteins than Coomassie blue [17]. The Campbell publication, therefore, directly repudiates Michalak’s claim that the HACBP and CRT are identical proteins.

Notably, the Ca^2+^-binding properties of the HACBP and CRT are distinct. CRT binds 1 Ca^2+^, with a Kd value of about 0.1 μM [18,19], and about 30 mol of Ca^2+^, with a Kd value of 1–2 mM [19,20]. Scatchard analysis of Ca^2+^ binding by CRT reveals a biphasic plot, which indicates the presence of two classes of Ca^2+^-binding sites [16,19,21]. In contrast, the Scatchard plot presented by MacLennan [7], which spans from 1 μM to 500 μM Ca^2+^, reveals the binding of one Ca^2+^ and has a single slope consistent with a single class of Ca^2+^-binding sites (Kd = 2.5–4 μM). This is consistent with the binding of one Ca^2+^ (Kd = 4 μM) and the absence of low-affinity Ca^2+^ sites reported in MacLennan et al. [6]. The studies of Ca^2+^ binding by CRT were performed using the equilibrium dialysis method (discussed in ref. [22]). Therefore, the inability of the HACBP to be stained by Stains-all and its binding of only a single high-affinity Ca^2+^ firmly establish that the HACBP and CRT must be distinct proteins [2].

The Michalak laboratory reported using an antibody to the HACBP to clone CRT from a skeletal muscle library. According to Michalak, that antibody was generated against the HACBP protein that was purified using the MacLennan rabbit skeletal muscle SR preparation. Michalak used the HACBP protein to generate antibodies that were then used to clone CRT from a skeletal muscle library. However, it was published in 1993 that a highly sensitive antibody to CRT failed to detect CRT in skeletal muscle SR preparations [23]. The inability to detect the HACBP/CRT in the SR is, therefore, consistent with reports from multiple laboratories using various techniques, including high-resolution SDS-PAGE followed by Coomassie blue or Stains-all staining, ^45^Ca^2+^ autoradiography, immunodetection, or mass spectroscopy, which have all failed to detect the HACBP or CRT in the skeletal muscle SR (Table 1).

**Table 1 cancers-17-00288-t001:** Analysis of Skeletal Muscle SR Proteins.

Study	Study Year	CSQ	HACBP	53 kDa Glycoprotein	Method of Detection	Reference
Ostwald and MacLennan	1974	+	+	ND	Coomassie Blue Stain ^1^	[13]
Michalak et al.	1980	+	+	+	Coomassie Blue Stain ^2,3^	[3]
Campbell et al.	1980	+	+ ^4^	+	Coomassie Blue Stain ^3^	[12]
Campbell and MacLennan	1981	+	ND	+	Coomassie Blue Stain ^3^	[4]
Campbell and MacLennan	1982	+	ND	+	Coomassie Blue Stain ^3^	[24]
Campbell et al.	1983	+	ND	+	Coomassie Blue Stain ^2^, Stains-All	[25]
Campbell et al.	1983	+	ND	+	Coomassie Blue Stain ^2^, Stains-All	[17]
Macer and Koch	1988	+	ND	+	Coomassie Blue Stain ^2^, ^45^Ca^2+^ Autoradiography	[26]
Damiani et al.	1988	+	ND		^45^Ca^2+^ Autoradiography	[27]
Leberer et al.	1989	+	ND	+	Coomassie Blue Stain ^2^, ^45^Ca^2+^ Autoradiography	[28]
Hofmann et al.	1989	+	ND	+	^45^Ca^2+^ Autoradiography, Stains-All	[29]
Van et al.	1989	+	ND	ND	Stains-All	[16]
Leberer et al.	1990	+	ND	+	Coomassie Blue Stain ^2^	[30]
Cala et al.	1990	+	ND	ND	Stains-All	[31]
Damiani and Margret	1991	+	ND	+	Stains-All, ^45^Ca^2+^ Autoradiography, Ponceau Red ^3^	[32]
Raeymaekers et al.	1993	ND	ND	ND	Immunodetection with CRT Antibody	[23]
Treves	2009	+	ND	+	Coomassie Blue	[33]
Staunton	2012	+	ND	+	Mass Spectrometry	[34]

**+**, protein detected; ND, not detected. This table summarizes the data presented by multiple laboratories, detailing the proteins of the skeletal muscle SR. ^1^ the method of SDS-PAGE was according to Weber and Osborn [9]. MacLennan reported that in the SDS-PAGE analysis of the rabbit muscle, the skeletal muscle SR yielded several Coomassie blue staining bands, including a 55 kDa band that he called the HACBP. ^2^ analysis by 2-D gel—Weber and Osborn (first dimension) and Laemmli (second dimension). Michalak reported that the 55 kDa band (HACBP) detected using Weber and Osborn SDS-PAGE resolved into five distinct bands [11]. ^3^ SDS-PAGE, according to Laemmli. ^4^ Campbell reported the 55 kDa band (HACBP) detected using Weber and Osborn SDS-PAGE resolved into 53 kDa, 55 kDa, and 56 kDa bands [12]. The 53 kDa band (a 53 kDa glycoprotein) comprised greater than 90% of the staining intensity. None of these bands reacted to Stains-All [5].

### 2.2. Does the HACBP Actually Exist?

To rationalize the existence of the HACBP, we must test two possible scenarios. The first scenario is that MacLennan, as stated in his several publications published around 1974 [6,7,13,14,35], purified 1 mg of the HACBP from 800 g of rabbit skeletal muscle as a single pure protein. The second scenario is that MacLennan purified 1 mg of a mixture of three to five 55 kDa proteins from 800 g of rabbit skeletal muscle (equivalent to two rabbits). If we test scenario 1, we must assume that MacLennan purified about 1 mg of pure HACBP protein and not a mixture of proteins from the skeletal muscle SRs of two rabbits. To avoid confusion as to which of the multiple bands of the 55 kDa Coomassie blue staining band is the HACBP, we will utilize Campbell’s analysis and refer to the HACBP as HACBP-2. This scenario would identify HACBP-2 as an abundant protein of the rabbit skeletal muscle SR, which would be easily detectable by the Coomassie blue staining of SDS-PAGE gels. Let us consider that the 55 kDa band comprises more than 90% 53 kDa glycoprotein and only 5% HACBP-2. We can back-calculate the total protein content of the 55 kDa band obtained from the skeletal muscle SRs of two rabbits, using relative ratios and assuming 1 mg of pure HACBP-2. This calculation suggests that about 18 mg of the 53 kDa glycoprotein, 1 mg of HACBP-2, and 1 mg of other proteins (pyruvate kinase, etc.) comprise the HACBP band, i.e., about 20 mg in total. Thus, comparing the 55 kDa band (20 mg) with that of calsequestrin (6 mg) would suggest that the 55 kDa proteins are collectively more prevalent than calsequestrin by threefold. Therefore, in Weber and Osborn SDS-PAGE, the Coomassie-blue-stained 55 kDa band should be more intense than the calsequestrin band. However, calsequestrin staining is actually at least twice as intense as HACBP staining [7]. Also, according to this scenario, the HACBP should be easily detectable, but, clearly, this is not the case; see (Figure 1) and Table 1.

The second scenario is that MacLennan purified 1 mg of a mixture of several 55 kDa proteins. We believe this is the true scenario. MacLennan stated, “Recently, we reexamined the contribution of the high-affinity calcium-binding protein to the 55,000-Dalton band, using the Laemmli slab gel system and a two-dimensional slab gel system utilizing a Weber–Osborn separation in one dimension and a Laemmli system in the second dimension. We found some four-protein bands in the 55,000-Dalton region, the most prominent of which was a diffusely staining protein of about 53,000 Daltons” [10]. Unfortunately, MacLennan did not identify, nor has ever identified, which, if any, of the four bands was the HACBP. This scenario would mean that the characterization of this protein mixture as a single protein, published by MacLennan [3,6,7,10,13,14], is erroneous. If we assume that the HACBP-2 is one of the five proteins detected by Michalak [3], which MacLennan pooled and called the HACBP; then, considering that 90% of this band is the 53 kDa glycoprotein and 2% is pyruvate kinase, about 5% is expected to be the HACBP-2. According to MacLennan, 1 mg of the 55 kDa protein band was obtained from 800 g of rabbit muscle, corresponding to two rabbits. Therefore, about (5% of 1 mg) 0.05 mg of pure HACBP would be recovered from two rabbits. This scenario would be consistent with the SDS-PAGE analysis of the SR preparation (Figure 1). It would also be consistent with the fact that the HACBP-2 is only a minor protein of the SR. However, this scenario creates another issue. The antibody generation strategy, used by Michalak to prepare a CRT antibody for use in his detection of CRT clones in a λgt11 library, required the injection of 3 mg of protein into a goat. Michalak claimed to purify the HACBP to prepare antibodies, using the MacLennan procedure [36]. Assuming that two goats were used, that would mean that (6/0.05 × 2) or 240 rabbits would be required for sufficient purified HACBP to inject and produce the antibodies used to clone CRT [1].

### 2.3. Are the HACBP and CRT the Same Protein?

Regardless of whether or not the HACBP exists as a pure protein or as a mixture of multiple proteins, the fact is that the HACBP does not react with Stains-all (Figure 1 and Figure 2) and has markedly different Ca^2+^-binding properties than CRT, especially the absence of multiple low-affinity Ca^2+^/Mg^2+^ sites [6,7]. Therefore, CRT and the HACBP are distinct proteins. Furthermore, our extensive analysis has concluded that more than ten laboratories have not detected CRT in the skeletal muscle SR (Table 1). Therefore, we must conclude that CRT is absent in the rabbit skeletal muscle SR, unlike the HACBP.

### 2.4. Discovery of CRT as Calregulin

We propose that CRT was initially discovered not as the HACBP but as calregulin, a major Ca^2+^-binding protein of the liver ER, in 1984 and 1985 [37,38], and partially sequenced in 1987 [39]. Subsequently, Van et al. [16] identified calregulin by N-terminal sequencing analysis of the ERs of several rat tissues. They confirmed that calregulin bound 1 mole of Ca^2+^ per mole of protein (Kd = 1 μM) in the presence of Mg^2+^ and was stained by Stains-all, and, importantly, they were unable to detect calregulin in the skeletal muscle SR by Stains-all staining, whereas calsequestrin was easily visualized (Figure 2). In 1989, calregulin was renamed as calreticulin by Koch, MacLennan, and Michalak [40].

## 3. Calreticulin–Gene Structure

The human Ro/SS-A autoantigen was shown to match the N-terminal sequence of calregulin [41,42] and, later, the entire sequence of CRT [40]. The human Ro/SS-A autoantigen (CRT) was localized on chromosome 19 at locus p13.3–p13.2 [41,43]. The CRT promoter regions consist of binding sites for transcription factors, such as NKx2.5 (NK2 transcription factor related, locus 5), COUP-TF1 (chick ovalbumin upstream promoter transcription factor 1) [44], MEF2C (myocyte enhancer factor-2C) [45], GATA6 (GATA binding protein 6), Evi-1 (ecotropic viral integration site-1) [46], and OLF-1 (olfactory-neuron-specific transcription factor) [47], and has retinoblastoma control elements [48]. Calcium depletion from the ER and ER stress are key activators of CRT expression [49,50,51].

## 4. CRT Domain Structure

CRT has a molecular mass of 46.9 kDa and an anomalous apparent molecular mass, in SDS-PAGE, of 63 kDa [37]. As discussed, CRT was originally called calregulin and was shown to localize in the ER [18,21,37,38,39,52]. Calregulin was shown to localize in the Ca^2+^ storage regions of the ER [16].

CRT is synthesized as a precursor protein containing an amino terminally located signal peptide (residues 1–17). CRT is composed of three structural and functional domains: a highly conserved amino-terminal N-domain (residues 18–197), a central proline-rich P-domain (residues 198–308), and a carboxyl-terminal C-domain (residues 309–417) [40,53]. The P- and C-domains extend from the N-domain core, and in the presence of Ca^2+^, the P-domain hairpin-like structure folds back on itself and associates with the C-domain.

Like its ER membrane homolog, calnexin, CRT has an amino-terminal N-domain that is a highly conserved, stable globular structure with eight antiparallel β-strands and a disulfide bridge [54,55,56]. The N-domain possesses low-affinity, high-capacity, Zn^+^-binding sites (Kd = 310 μM and 14 moles of zinc per mole of CRT) [21,57] and a binding site for adenosine triphosphate (ATP) [58]. Although initially reported to be in the P-domain [53], the N-domain also has a single Ca^2+^-binding site, with a sub-micromolar Kd value [16,18,21,38]. This Ca^2+^ is coordinated by seven oxygen atoms contributed by the bidentate side chain of Asp328; backbone carbonyls of Gln26, Lys62, and Lys64; and two water molecules [56]. The high-affinity Ca^2+^-binding site is conceptually similar to another well-described Ca^2+^-binding site, namely, the endonexin fold of the annexin proteins. The endonexin fold is considered to be the signature amino acid sequence present in the annexin family of Ca^2+^-binding proteins and houses the Ca^2+^-binding motif (KGXGT-38 residues—D/E) (reviewed in [59]). However, these sites are Ca^2+^/Mg^2+^ sites, as although Mg^2+^ does not activate phospholipid binding, it does inhibit Ca^2+^-dependent phospholipid binding [60].

The N-domain of CRT possesses polypeptide- and carbohydrate-binding sites [58,61,62,63,64]. CRT exerts this chaperone function by recognizing monoglucosylated N-linked glycans (Glc_1_Man_5–9_GlcNAc_2_) present transiently on nascent glycoproteins in the ER [63,65].

The polypeptide binding site consists of mainly hydrophobic interactions provided by Phe74, Trp319, Cys105, Cys137, and Asp135 [55]. The lectin binding region can bind a tetrasaccharide via an extensive network of hydrogen bonds and hydrophobic contacts. The oxygen O(2) in glucose occupies a pocket formed by CRT side chains while forming direct hydrogen bonds with the carbonyl of Gly(124) and the side chain of Lys(111). In contrast, the Cys(105)-Cys(137) disulfide bond interacts with the third and fourth sugar moieties of the Glc(1)Man(3) tetrasaccharide. The chaperone activity of CRT is induced by various conditions associated with ER stress, including calcium depletion and heat shock [66]. Interestingly, Ca^2+^ is essential for CRT to bind isolated oligosaccharides and to recognize glycosylated proteins in vitro [63,65], whereas Zn^2+^ stimulates the ability of CRT to suppress the thermal aggregation of unglycosylated proteins in vitro [67]. Mutagenesis studies have suggested that the lectin-binding site is the predominant contributor to the chaperone function of CRT [68]. However, CRT can bind to denatured proteins independent of their glycosylated status [69]. This interaction is dependent on both Ca^2+^ and ATP.

It was originally proposed that the N-domain also interacts with the DNA-binding site of glucocorticoid receptors and prevents the receptors from binding to their specific glucocorticoid response element [70]. However, this claim was later rescinded [71,72].

The central P-domain of CRT is high in proline content and possesses several regions consisting of repeated amino acid sequences. The repeated motif 1 (I-DPD/EA-KPEDWDD/E) and motif 2 (G-W- -P-I-NP-Y) are present as 1,1,1,2,2,2. The P-domain also forms a relatively rare β-stranded hairpin-like structure formed by the proline-rich region. The hairpin-like structure forms a hook-like arm, the tip of which contains the binding site for ERp57 [73]. This region is stabilized by a three-strand antiparallel β-sheet formed by the amino acid repeat sequences. The P-domain interacts with other proteins that function as ER chaperones and form ER chaperone–substrate complexes, such as ERp29 [74], CypB [75], PDI1A [76,77,78], and ERp57 [61,79,80]. This region, along with the central N-domain of the protein, participates in the chaperone function of CRT. Although initial studies localized the high-affinity Ca^2+^ site to the P-domain [53], more recent studies have established that the N-domain contains this site [67,81].

The carboxyl-terminal C-domain of CRT contains many acidic amino acid residues that are responsible for forming multiple low-affinity Ca^2+^/Mg^2+^-binding sites. These sites bind about 30–50 moles of Ca^2+^ per mole of protein, with a Kd value of 1–4 mM [20,82]. The low-affinity sites are not specific for only Ca^2+^ and are not detected in the presence of millimolar Mg^2+^ [16,21]. The CRT C-domain displays a disordered structure at low Ca^2+^ concentrations, whereas at higher Ca^2+^ concentrations, it assumes a more rigid and compact conformation [83]. The last four residues of the C-domain possess the sequence KDEL, which functions as an ER lumen retention/retrieval domain [84]. However, two independently operating retention/retrieval mechanisms have been suggested for CRT based on mutagenesis studies: one is a Ca^2+^-dependent mechanism providing for direct retention in the ER and involving the C-domain; the other is a KDEL-based retrieval system for escaped CRT present in the Golgi apparatus [85]. Furthermore, the deletion of the C-domain does not affect the chaperone activity of CRT, suggesting that the main roles of CRT in the ER, lectin chaperone, and Ca^2+^ buffer are mutually independent [86]. The C-domain’s Ca^2+^/Mg^2+^ sites also mediate the interaction of extracellular CRT with the plasma membrane’s phosphatidylserine and in CRT-mediated phagocytosis [87]. The domain structure of CRT is presented in Figure 3.

**Figure 3 cancers-17-00288-f003:**
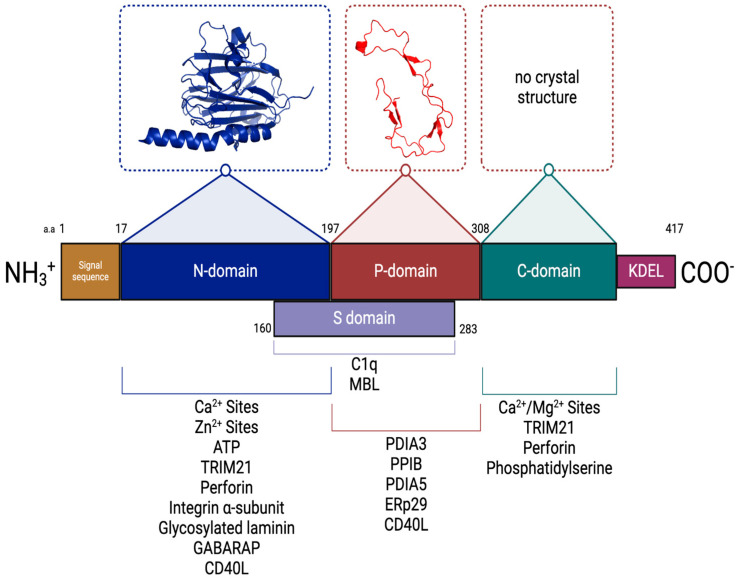
Structure of calreticulin domains. The three-dimensional models are based on NMR, X-ray crystallography, and Cryo-EM data. The N-domain (PDB ID:3POW) and P-domain (PDB ID:1HHN) are shown. A crystal structure is not available for the C-domain. The calreticulin protein consists of a signal peptide (tan), an N-terminal globular domain (blue), a central proline-rich P-domain (red), a C-terminal Ca^2+^/Mg^2+^-binding C-domain (green), and an endoplasmic reticulum retention signal, KDEL (purple). A subdomain (S-domain) is located between amino acids 160 and 283 [88,89]. Binding partners of CRT (summarized in Table 2) are shown below their interacting domain; a.a. (amino acids).

**Table 2 cancers-17-00288-t002:** Interaction partners of calreticulin.

Protein	Binding Affinity (Kd)	CRT Domain	Reference(s)
TRIM21 (tripartite motif-containing protein 21)	NR	N- and C-terminal domains	[90]
PDIA3 (protein disulfide–isomerase A3, ERp57)	5–9 μM	P-domain	[80,91,92]
Perforin	1.2 μM	N- and C-domains	[93,94,95]
Integrin α-subunit	0.25 μM	N-domain	[96,97]
PPIB (peptidyl–prolyl cis–trans isomerase B)	10 μM	P-domain	[75]
PDIA5 (protein disulfide–isomerase A5)	0.16 μM	P-domain	[98]
Glycosylated laminin	0.5 μM	N-domain	[99]
GABARAP (Gamma-aminobutyric acid receptor-associated protein)	64 nM	N-domain	[100,101]
SPACA9 (sperm acrosome-associated protein 9)	NR	NR	[102]
CLCC1 (chloride channel CLIC-like 1)	NR	NR	[103]
C1q	100 nM	N- and P-domains	[88,104,105,106]
MBL (mannose-binding lectin)	NR	N- and P-domains	[107]
ERp29	13 μM	P-domain	[74]
CD40L	50 nM	N- and P-domains	[108]
LDL receptor-related protein 1 (CD-91)	NR		[109]
Thrombospondin (TSP-1)	NR	N-domain	[110,111]
Phosphatidylserine	12 μM	C-domain	[87]

The data for documented binding partners of CRT are included in this table. CRT also interacts with many ER proteins and transiently associates with numerous different newly synthesized glycoproteins as they enter the secretory pathway. These interactions are not included. NR, not reported. The C1q- and MBL-binding sites are present in a subdomain contributed by the N- and P-domains, called the S-domain (residues 160–283) [88,89].

## 5. CRT–Metal Ion Binding Properties

The initial studies of the cation-binding properties of CRT suggested that the high-affinity Ca^2+^-binding site and the Zn^2+^-binding sites in CRT were relatively specific. The high-affinity Ca^2+^ binding is unaffected by a 500-fold excess of Zn^2+^, Mg^2+^, Mn^2+^, or Fe^2+^, whereas Zn^2+^ binding, as monitored by changes in intrinsic fluorescence, was unaffected by a 10-fold excess of Ca^2+^, Mg^2+^, or Mn^2+^ [16,21]. We initially observed that Ca^2+^ and Zn^2+^ induce different conformational changes in CRT. Ca^2+^ binding results in the movement of tryptophan away from the solvent, whereas Zn^2+^ causes a movement of tryptophan into the solvent and the exposure of a hydrophobic region [21]. Subsequently, it was shown that the occupancy of the high-affinity Ca^2+^ site in CRT increases the protein’s thermal stability and conformational rigidity as a consequence of an increase in hydrophobic packing but does not change its overall three-dimensional structure [67]. In contrast, the occupancy of the low-affinity Ca^2+^ sites increases the protein’s thermal stability and conformational rigidity, also because of an increase in the stabilization of the hydrophobic core [67]. The occupancy of the low-affinity zinc sites alters the internal hydrophobic packing of the protein, resulting in a more expanded tertiary structure.

Studies of the best-described Ca^2+^-binding motif, the “EF-hand” [112], have allowed the classification of Ca^2+^-binding sites, according to their specificity toward Mg^2+^, into specific Ca^2+^ sites and Ca^2+^/Mg^2+^ sites (reviewed in [113,114]). For example, Li et al. [67] have shown that CRT is proteolyzed by chymotrypsin in the presence of 3 mM MgCl_2_. However, the addition of micromolar Ca^2+^ in the presence of Mg^2+^ confers complete resistance to digestion by chymotrypsin. This further suggests that Mg^2+^ does not compete with Ca^2+^ for binding to the high-affinity Ca^2+^ site.

To understand the functional significance of the Ca^2+^- and Zn^2+^-binding sites, it is necessary to appreciate the physiological concentrations of Ca^2+^, Zn^2+^, and Mg^2+^ in the cell. The total (free and bound) Ca^2+^ concentration within the ER is estimated to be about 2 mM, while the free ER Ca^2+^ varies from 50 to 500 μM [115,116]. The ER is one of the major cellular Mg^2+^ pools, with a total concentration from 14 to 18 mM [117] and free concentrations from 0.5 to 1 mM [118]. In contrast, free Zn^2+^ in the ER is about 5 nM [119]. Zn^2+^ is required for ER functions, and its deficiency can lead to an increase in the level of unfolded protein and ER stress because many proteins bind or acquire zinc in the ER [120]. Regarding the interactions of Ca^2+^ and Mg^2+^ with a Ca^2+^-binding site, multiple studies have shown that the conformational changes in proteins, resulting from Ca^2+^ or Mg^2+^ binding, are distinct. Differences in ionic radii, hydration, and coordination geometry are essential properties that enable Ca^2+^-binding proteins to discriminate between Ca^2+^ and Mg^2+^ [121]. Most importantly, Ca^2+^/Mg^2+^-binding sites typically display unique conformational changes in response to Ca^2+^ or Mg^2+^ because Mg^2+^ is much smaller than Ca^2+^. The coordination of Ca^2+^ and Mg^2+^ by proteins is also different [122], which explains why Ca^2+^ and Mg^2+^ produce distinct conformational changes in a protein.

The Ca^2+^-binding sites in calmodulin are called “EF-hand” motifs. The EF-hands are helix–loop–helix binding motifs and consist of residues at positions 1 (+x), 3 (+y), 5 (+z), 7 (−y), 9 (−x), and 12 (−z), which are arranged in a pentagonal bipyramid, allowing for the coordination of metal cations [123]. Studies with “EF-hand”-containing proteins have established two distinct types of Ca^2+^-binding sites. The first type, the specific Ca^2+^ sites, typically bind Ca^2+^ in the presence of millimolar Mg^2+^. The second, the Ca^2+^/Mg^2+^ sites bind both Ca^2+^ and Mg^2+^ and Ca^2+^-binding at micromolar concentration is usually not detected in the presence of millimolar Mg^2+^. It has been shown that Mg^2+^ may be an important factor in modulating the Ca^2+^ sensitivity of proteins, particularly switching off Ca^2+^-regulated systems at resting cellular Ca^2+^ levels [121]. It is, therefore, likely that the Ca^2+^/Mg^2+^ sites in CRT do not buffer significant Ca^2+^ at resting ER Ca^2+^ levels but may be activated during Ca^2+^ influx into the ER. Consistent with that suggestion is our observation that CRT binds only 1 mole of Ca^2+^ per mole of CRT in the presence of millimolar Mg^2+^ [38,39]. This has also been observed by others [81]. This is also consistent with a report by Van de Putt, who reported that CRT could only be responsible for the binding of 3–4% of the total Ca^2+^ stored in the ERs of pancreatic acinar cells [124].

Within the prevailing ER Zn^2+^ concentrations, CRT is unlikely to bind appreciable amounts of Zn^2+^. However, it is possible that the binding of CRT to ER proteins could drastically influence the affinity of Ca^2+^, and possibly Zn^2+^, as has been observed for the Ca^2+^-binding protein annexin A2 [125]. In this example, annexin A2 binds Ca^2+^, with a millimolar Kd value, but micromolar concentrations are required for the Ca^2+^-dependent aggregation of chromaffin granules by annexin A2.

The functional significance of multiple cation-binding sites has been addressed by Li et al. [67]. They suggest that CRT may have the intrinsic ability to display two distinct metal-dependent chaperone functions: a high-affinity Ca^2+^-dependent lectin function or a more classical low-affinity Zn^2+^-dependent chaperone function.

## 6. Calreticulin Functions: From the Endoplasmic Reticulum to the Extracellular Surface

Since the discovery of CRT [37,38], CRT has been shown to be a multifunctional protein involved in over 40 intra- and extracellular processes in mammalian cells [126,127]. The established binding partners for CRT are presented in Table 2.

Key intracellular functions include roles in Ca^2+^ signaling and Ca^2+^ storage [128,129,130], a role in MHC class I assembly [131,132], and a role as a chaperone that functions to promote the correct folding of newly synthesized proteins in the ER [58,115,129,133]. CRT is also present in the cytoplasm, where it serves to shuttle specific proteins between the ER and Golgi apparatus [115], and in the nucleus, where it mediates the export of many nuclear receptors, such as the glucocorticoid receptor [134,135]. Chemotherapeutic agents, such as anthracyclines and oxaliplatin, provoke CRT translocation to the extracellular surface [92,136]. Extracellular CRT is a part of a damage-associated molecular pattern (DAMP) complex, which plays a role in the phagocytic response [137,138]. However, CD47, ubiquitously expressed in normal tissue, is highly expressed by a wide range of cancer cells [139] and blocks the phagocytosis of CRT-expressing cancer cells. The proposed functions for CRT are summarized in Table 3.

**Table 3 cancers-17-00288-t003:** Calreticulin functions.

Function	References
Protein Folding	[58,63,64,115,133]
Calcium Homeostasis	[124,140,141,142]
Cell Adhesion	[99,111,143,144,145]
Wound Healing	[146,147]
Phagocytosis	[107,138,148,149]
Immunogenic Cell Death	[148,150,151,152]
RNA Stability	[153,154]
Plasminogen Receptor	[155]

Listed are the well-documented functions of CRT.

Studies with CRT-null mice have shown that the loss of CRT is embryonically lethal [156,157,158]. Dedhar’s group initially reported that CRT-null mice die in utero on days 14–14.5, primarily because of defects in cardiac morphogenesis [158]. This was confirmed by Michalak [156]. Dedhar further reported that CRT is not essential during the early stages of embryonic development but is important for the development of the heart and brain and for ventral body wall closure [145]. Dedhar has suggested that the reported abnormalities caused by the deletion of CRT to be because of the role of CRT in the modulation of cellular Ca^2+^ signaling [145]. Similarly, mice that overexpress cardiac CRT show severe cardiac pathology and sudden death [159].

### 6.1. CRT and Ca^2+^ Homeostasis

The ER is crucial for Ca^2+^ storage and intracellular Ca^2+^-mediated signaling [160,161], and its dysregulation in diseases has been thoroughly reviewed [162,163,164]. The dysregulation of Ca^2+^ homeostasis is an important feature in cancer development (reviewed in [165]). The ER maintains a steep Ca^2+^ gradient between its lumen (from 100 μM up to 1 mM [Ca^2+^]) and cytosol (~100 μM [Ca^2+^]) [166,167] through various influx mechanisms, e.g., the sarcoplasmic/endoplasmic reticulum calcium ATPase pump (SERCA) [168], and various efflux mechanisms, e.g., inositol 1,4,5 triphosphate [169], ryanodine receptors [170,171], and Sec61 translocation [172]. Ca^2+^ buffering is maintained by several proteins, including CRT [20,82], through its negatively charged C-domain; BiP/Grp78 [173,174]; and Grp94 [16,26], which binds Ca^2+^ at both high- and low-affinity sites.

ER Ca^2+^ stores are replenished by Ca^2+^-release-activated Ca^2+^ channels (CRACs), stromal interacting molecules (STIMs) [175,176], and calcium-release-activated calcium channel protein 1 (ORAI1) proteins [177,178,179]. STIM proteins transmit the Ca^2+^-storage-depletion message to the plasma membrane and participate in the refilling of the ER by physically interacting with the ORAI1 Ca^2+^ channels at the plasma membrane. A decrease in luminal-ER Ca^2+^ promotes conformational changes in the STIM1 via its ER luminal Ca^2+^-binding sites, resulting in oligomerization and the activation of ORAI1 channels and the formation of STIM1-ORAI1 membrane clusters [180]. The resultant influx of Ca^2+^ allows the filling of the ER by the SERCA pump. The oxidizing environment in the ER can activate the STIM1 and is also important for forming disulfide bonds in proteins and protein folding [181,182,183,184].

Along with CRT, BiP/GRP78 [173] and GRP94 [26] play crucial roles in Ca^2+^ storage and buffering. GRP94 undergoes conformational changes in response to Ca^2+^ binding, affecting its chaperone function [185]. BiP/GRP78 contains an ATPase site, and its Ca^2+^-binding properties are influenced by its interaction with ATP and adenosine diphosphate (ADP) [174].

Liu et al. were the first to propose a direct relationship between CRT and Ca^2+^ storage capacity [140]. In that study, the expression of CRT was inhibited by antisense RNA, and IP_3_-dependent Ca^2+^ transients were significantly reduced [140]. Subsequently, it was shown that overexpression of CRT inhibited repetitive IP_3_-induced Ca^2+^ waves. Deletion mutagenesis demonstrated that CRT inhibition of these waves was mediated by the high-affinity Ca^2+^ site [128]. Subsequently, in an elaborate series of experiments, Bastianutto et al. [141] transfected HeLa cells with an approximately 3.5-fold overexpression of CRT and observed that CRT participated in Ca^2+^ buffering within the IP3-sensitive Ca^2+^ stores and contributed to about 45% of these stores. They also made several fundamental observations, including that the free Ca^2+^ within the lumen of the IP_3_-sensitive stores was in the millimolar range and that Ca^2+^ influx across the plasma membrane, activated by depletion of the stores, was directly dependent on the ER lumenal Ca^2+^ concentration. Overexpression of CRT was also shown to inhibit repetitive IP_3_-induced Ca^2+^ waves [128]. Mery et al. [186] overexpressed CRT by 1.6-fold in fibroblasts, observed a 2-fold increase in cellular Ca^2+^, and demonstrated that 80% of the increased Ca^2+^ content was found within thapsigargin-sensitive Ca^2+^ stores. IP_3_ causes Ca^2+^ release, resulting in oscillations and waves in *Xenopus laevis*. These oscillations result from the opening of the IP_3_-bound IP_3_ receptor, which is activated by low Ca^2+^ concentrations but inhibited by high Ca^2+^ concentrations. The SERCA pump can remove the inhibitory effect of a high cytosolic Ca^2+^ concentration on the IP_3_ receptor by pumping cytosolic Ca^2+^ into the ER. John et al. [187] observed that CRT interacted with the C-terminus of SERCA and regulated its activity. CRT was also reported to regulate IP_3_-mediated Ca^2+^ release and capacitative Ca^2+^ entry in HeLa cells [188], consistent with the Xenopus studies. Furthermore, Xu et al. [189] showed that the C-domain of CRT mediated the ability of CRT to reduce increases in cytosolic Ca^2+^ because of Ca^2+^ influx and for CRT to attenuate the InsP_3_-induced decline in the free-Ca^2+^ concentration within the ER lumen.

Kwon et al. [142] showed that CRT is associated with the integrin cytoplasmic domain and mediated the coupling between the Ca^2+^ release and Ca^2+^ influx (Figure 4). They suggested that CRT served as a cytosolic integrin activator and a signal transducer between integrins and Ca^2+^ channels on the cell surface (Figure 4). Coppolino et al. [145] showed that the resting cytosolic Ca^2+^ levels in wildtype and CRT-null mouse embryonic fibroblasts were similar, and the cytosolic Ca^2+^ transient elicited by the addition of thapsigargin to cells, which is a measure of the Ca^2+^ stored in the ER, was indistinguishable. However, they did observe that the absence of CRT perturbed the integrin-mediated influx of extracellular Ca^2+^ (Figure 4). They suggested that the proposed role of CRT in ER Ca^2+^ storage was not essential. Similarly, as discussed, Van de Putt reported that CRT could only be responsible for the binding of 3–4% of the total Ca^2+^ stored in the ERs of pancreatic acinar cells [124].

Michalak used CRT-null mouse embryonic fibroblasts to further explore the role of CRT as a regulator of Ca^2+^ homeostasis. They reported that the ERs of CRT-null fibroblasts have a lower capacity for Ca^2+^ storage, although the free ER luminal Ca^2+^ concentration was unchanged. CRT-null cells showed an inhibited Ca^2+^ release in response to agonist stimulation by bradykinin; however, IP_3_-dependent release was unchanged [129]. In contrast, fibroblasts from CRT-null mice showed no differences in ER Ca^2+^ storage from wildtype cells [145]. Recent studies have shown that the CRT function in ER Ca^2+^ storage is because of its regulation of PDI. Upon Ca^2+^ depletion, the reductive power of the cochaperone, ERdJ5, increases in part because of the sequestration of the oxidizing protein PDI. This reductive power is required to activate SERCA. Therefore, the CRT-PDI complex prevents the oxidation of ERdJ5, which regulates Ca^2+^ influx by reducing the disulfide bonds of SERCA, thereby activating SERCA, which pumps Ca^2+^ into the ER [78,190].

### 6.2. CRT and Integrin Binding

CRT plays a multidimensional role in integrin binding and function, directly affecting several cellular processes important for cancer progression, such as adhesion and metastasis. Early studies using CRT-null embryonic fibroblasts showed that integrin-mediated adhesion is drastically impaired despite normal integrin expression in these cells [145]. CRT is associated with the cytoplasmic domain of α-subunits of integrins (Figure 4) [96,191]. CRT interacts with the KXGFFKR sequence in the C-domain of the integrin α-subunit [96], which is crucial for maintaining the activation state of integrin and promoting integrin-mediated interaction with the extracellular matrix (ECM). Consistently, CRT-deficient fibroblasts and embryonic stem cells have impaired integrin-mediated adhesion to the ECM.

### 6.3. CRT Chaperone Function

Helenus originally proposed a model to explain how N-linked oligosaccharides affect glycoprotein folding in the ER [192]. Since then, the model has been extended (reviewed in [193]). Essentially, nascent polypeptide chains of the secretory pathway are translocated into the lumen of the ER, where they are rapidly N-glycosylated with Glc3Man9GlcNAc2 on asparagine residues in Asn-X-Ser/Thr motifs. Glucosidases then act on the polypeptide chain, and the monoglucosylated polypeptide chain specifically binds to the CRT (or calnexin) in the ER lumen. The nascent protein bound to CRT interacts with other chaperones to promote proper folding. The three key chaperones that interact with CRT are the oxidoreductase ERp57; its homologous relative, ERp29 [74]; and the peptidyl–prolyl cis/trans isomerase, CyB [80]. The glucose and three adjacent mannose residues form a carbohydrate recognition site for CRT. The remaining glucose is released by glucosidases, which release the polypeptide chain from CRT. Glycoproteins that are not properly folded can then be reglucosylated by the folding sensor of UDP-Glc, namely, the glycoprotein glucosyltransferase (UGGT), thereby redirecting them back to CRT binding or trimming by glucosidase II. Polypeptide chains that are correctly folded are trimmed by mannosidases and directed for trafficking out of the ER. Proteins that are terminally misfolded are trimmed by mannosidases and directed for proteolysis. High-affinity Ca^2+^ binding by CRT is required for nucleotide binding and chaperone stabilization [194]. As discussed, the N-terminal domain contains sites for binding to carbohydrates located on the polypeptide chain and regulatory co-factors, such as ATP and Ca^2+^.

CypB is associated with the P-domain of CRT at a site that appears to overlap with the ERp57-binding site [56]. The interaction of CRT with ERp57 is Ca^2+^ dependent, although a Zn^2+^-dependent binding was also suggested [76]. Crystallographic analysis has confirmed the interactions of ERp29, ERp57, and CypB with unique sites in the P-domain of CRT [195]. They showed that the ERp29-binding site was alpha-helical, while the CypB site was primarily composed of loops, and the ERp57 site was composed of one alpha-helix and two loops.

Corbett et al. have proposed a model to examine the Ca^2+^ dependence of CRT/chaperone association [77,196]. At low Ca^2+^ concentrations, CRT is associated with both protein disulfide isomerase (PDI) and ERp57 but not with the carbohydrates of newly synthesized glycoproteins. When Ca^2+^ is taken up by SERCA and the ER lumenal concentration is increased above 400 μm, Ca^2+^ binds to the C-domain of calreticulin, and the CRT dissociates from the PDI. Newly synthesized glycoproteins can now associate with calreticulin, and the formation of CRT-ERp57 complexes is enhanced, allowing accelerated chaperoning of glycoproteins.

### 6.4. CRT in MHC Class I Assembly and Antigen Presentation

MHC class I (MHC I) peptide loading and assembly are typically seen in virally infected cells and tumor cells. This process further mediates the immune recognition of virally infected target and cancer cells by cytotoxic T-cells and natural killer (NK) cells for subsequent killing. The assembly, peptide loading, and transport of MHC I is a complex process and involves multiple steps mediated by the peptide-loading complex (PLC). CRT is one of the components of the PLC, which includes other proteins, such as tapasin and Erp57, and transporters associated with antigen processing (TAP) (Figure 4) [197].

The assembly of MHC I begins in the ER, where newly synthesized MHC I heavy chains are associated with calnexin [198]. This is followed by binding the heavy chains to β2-microglobulin (β2m) to form a heterodimer. Subsequently, this heterodimer is loaded onto the PLC. The peptides to be loaded onto MHC I are generated in the cytosol by the proteasome. They are transported to the ER lumen and then to the PLC via the TAP protein. Once in the lumen, the N-extended peptides are cleaved by ER aminopeptidase (ERAAP) to optimize their length for effective binding to MHC I [199]. Tapasin then plays a key role in peptide loading onto MHC I. Tapasin plays a role as the MHC I editor of the PLC and can stabilize the empty MHC I heterodimer and promote the peptide loading of MHC I [200]. Overall, the PLC is crucial for every assembly step and ensures that only high-affinity peptides are loaded onto MHC I. Once the peptide is loaded, the MHC I–peptide complex dissociates from TAP and is transported from the ER to the cell surface via the Golgi apparatus and vesicles [201].

The multiple functions of CRT and their individual structural domains in MHC I assembly and transport have been elucidated using CRT-null fibroblasts [132]. Many studies have shown that a deficiency of CRT in cells results in reduced expression of MHC I on the cell surface. The reconstitution of wildtype CRT in CRT-deficient cells restored the cell surface expression of MHC I [132]. However, the CRT MPN C-terminal mutants failed to restore the expression of the surface’s MHC I [202], the functional liabilities of which are described later. Similarly, both the NP- and P-domains failed to rescue the expression of the surface’s MHC I in CRT-deficient cells and further suppressed the cell surface’s MHC I [203]. In the absence of CRT, the PLC does not function effectively, resulting in MHC I molecules binding to low-affinity peptides [202]. The globular domain of CRT utilizes ATP to release MHC I from the PLC. Briefly, the interaction of ATP and CRT destabilizes the globular domain of CRT, which further releases MHC I [194].

Those studies suggest that CRT is involved in the later stages of MHC I assembly. It acts as a chaperone, binds to partially folded MHC I heavy chains via the glycan in MHC I [63,131,204], stabilizes them, and prevents degradation. The association of CRT with the initial TAP–tapasin complex results in the loss of calnexin from the complex [198]. CRT facilitates the incorporation of the MHC I-β2m heterodimer into the peptide-loading complex. It interacts with other proteins, such as tapasin and Erp57, to further stabilize the complex [205,206]. CRT also retains incompletely loaded MHC I in the ER and, thus, acts as a quality control protein, preventing the transport of unloaded MHC I to the plasma membrane (Figure 4) [207]. Those studies collectively demonstrated the vital roles of CRT in MHC I assembly, peptide loading, and quality control for antigen presentation. This is summarized in Figure 4.

### 6.5. CRT in Immunogenic Cell Death

CRT plays multiple roles in the phagocytosis of various cells, ranging from bacteria to cancer cells and aging neutrophils. It interacts with multiple cell types via several receptors and proteins and, thus, plays crucial roles in immunosurveillance, cancer cell clearance, and cellular homeostasis. CRT was initially shown to bind to C1q and mannan-binding protein on the cell surface [105]. Subsequently, it was shown that the CRT-C1q interaction is involved in the C1q-bridging function, which plays a key role in phagocytosis [208,209]. In addition, it has been proposed that the acidic C-domain of CRT anchors it to exposed phosphatidylserine on the surface of apoptotic cells via Ca^2+^-phosphatidylserine-bridged interactions. The arm-like P-domain of CRT is proposed to interact with receptors on the phagocyte, such as low-density lipoprotein-receptor-related protein 1 or C1q, allowing CRT to function as a prophagocytic signal and promote the uptake of the dying cell [87].

Studies have shown that CRT is secreted by activated microglia, which then bind to and decorate *E. coli*, which mediates their phagocytosis by microglial cells [210]. Because antibodies and other opsonins, such as complement proteins, do not cross the blood–brain barrier (BBB), CRT released by the local microglial cells acts as a local opsonin and plays a crucial role in limiting infections in the brain. That study, therefore, showed that the microglia secrete CRT in response to lipopolysaccharide (LPS) treatments (Figure 5).

The role of CRT as an “eat-me” signal was first established in fibroblasts and cancer cells [138,148]. Those studies were the first to show that CRT is exposed on the surfaces of apoptotic fibroblasts, neutrophils, and Jurkat T-cells. CRT binds phosphatidylserine on apoptotic cells, which initiates phagocytosis by phagocytic cells, such as macrophages [138]. Furthermore, CRT exposed on the surface of apoptotic cells interacts with low-density lipoprotein-receptor-related protein 1 (LRP1) in the phagocyte to mediate engulfment. In a subsequent study, Obeid et al. showed that CRT is rapidly transported to the cell surface in pre-apoptotic anthracycline-treated cancer cells. The cell surface’s CRT was later shown to be essential for the engulfment of the cancer cells by dendritic cells (DCs) to elicit an immune response (Figure 5B). This process, which causes cancer cell death by generating an immune response, is known as immunogenic cell death (ICD) [148]. Those studies showed, for the first time, the importance of the cell surface’s CRT in mediating the engulfment of cancer cells by phagocytes, such as DCs, and the anticancer immune response, or ICD, in vivo. In this case, the transport of CRT to the cell surface was inhibited by the depletion of ERp57, a molecular chaperone for CRT, and co-localized with CRT at the plasma membrane. ERp57, however, does not possess immunogenic properties [211,212].

Research by Panaretakis et al., in 2009, elucidated the mechanisms underlying pre-apoptotic CRT exposure in ICD [136]. They showed that three distinct signaling pathways, namely, ER stress with eIF2α phosphorylation, apoptosis-associated caspase-8 activation, and ER-to-Golgi secretion, worked concertedly to promote the SNARE-dependent exocytosis of CRT-containing vesicles. Interestingly, only ICD inducers activated all three pathways simultaneously. Further studies have shown that apart from chemotherapeutics, such as anthracyclines and oxaliplatin, radiation, photodynamic therapy, and oncolytic viruses are also effective in CRT cell-surface exposure and promoting antitumor immunity [213,214,215].

The role of CRT as an ICD inducer has an important clinical significance. Several studies in multiple cancers have shown that CRT exposure can predict the efficacy of anticancer therapies and patient outcomes [216,217,218]. Clinical monitoring of intratumoral and plasma levels of CRT has significant potential to predict the efficacy of immune-checkpoint-inhibitor therapies (ICI) and other chemotherapies. Imaging techniques targeting the cell surface (ecto-CRT) have been developed for early ICD detection and treatment response prediction in cancer patients [219].

CRT is a central player in macrophage-mediated programmed cell removal (PrCR). During cellular homeostasis, PCR is vital for removing unwanted, damaged, dysfunctional, and aged cells. Studies by Feng et al. showed that activated macrophages secrete CRT, which is bound on the surfaces of both viable and apoptotic neutrophils via asialoglycans. This further initiates their removal, thus contributing to reduced inflammation and the maintenance of tissue homeostasis [220].

In summary, the roles of the cell surface’s CRT in cell death and removal have implications for biological processes, such as cancer surveillance and neutrophil clearance (in both resolving inflammation and wound healing). This opens up potential therapeutic opportunities in wound healing and cancer treatment.

## 7. Calreticulin Mutants and Cancer: Impacts on Biological Functions

Essential thrombocytosis (ET) belongs to a sub-category of myeloproliferative neoplasms (MPNs) characterized by increased circulating platelets. Thrombotic and hemorrhagic complications are frequent in patients with ET, and their most significant health risk is an increased risk of developing blood clots [221,222,223,224]. It has been reported that ET patients have significant hypofibrinolysis and exhibit a 34% longer clot lysis time than matched controls [225]. Those studies presented the possibility that diminished plasmin generation could account for the hypofibrinolysis observed in patients with ET.

An appreciation of the significance of the role of hypofibrinolysis requires an understanding of the plasminogen/plasmin system. Plasminogen is an important blood protein and zymogen of the protease plasmin. Plasminogen activators, such as the urokinase-type plasminogen activator (uPA) and tissue plasminogen activator (tPA), are secreted by cells and convert plasminogen to plasmin. This activation results from the cleavage of an Arg^561^-Val^562^ peptide bond within the plasminogen, giving rise to the active protease, plasmin. Plasmin cleaves fibrin, the major component of blood clots and certain extracellular matrix proteins (reviewed in [226,227,228]). The role of plasmin in maintaining blood patency is well established, and the dysfunction of the plasmin fibrinolytic system has been linked to heart attacks and strokes [227,229,230]. The ability of the plasminogen activators to generate plasmin from plasminogen is minimal; however, when plasminogen is bound to cell-surface plasminogen receptors, this rate is drastically increased (reviewed in [231,232,233,234]). These plasminogen receptors play key roles in many physiological and pathological processes ranging from blood clot resolution to cancer invasion and metastasis and to host immunity [235].

Studies have found that mutations in three genes are responsible for ET. Specifically, Janus kinase 2 (JAK2), thrombopoietin receptor (MPL), and CRT mutations are mutually exclusive and account for 50–60% and 40–50% of ET, respectively. Whole-exome sequencing has identified recurrent mutations in CRT in 70% of ET patients without a JAK2 or MPL mutation [236,237].

CRT mutations in MPN patients were first reported in 2013, exclusively in subjects presenting with ET and MF. Specifically, in patients with wildtype JAK2, 70–84% had CRT mutations. Furthermore, this first study identified 148 CRT mutations with 19 distinct variations. The CRT mutations were localized to exon 9 and caused a +1 frameshift, resulting in a modified C-terminus of CALR. This frameshift causes either a 52 bp deletion (Type I) or a 4 bp insertion (Type II), which results in a highly positively charged C-terminus region compared to the negatively charged amino acids in the CRT wildtype protein. These mutations were observed in hematopoietic stem and progenitor cells in the ER, with no significant accumulation in the Golgi apparatus or at the cell surface. These and other studies have also reported that although patients with CRT mutations had higher numbers of platelets and lower leukocyte cell counts and hemoglobin and hematocrit levels, they also had a lower risk for thrombosis compared to the patients with JAK2 mutations [236,237,238,239,240]. Interestingly, thrombosis-free survival was significantly longer in patients harboring CRT mutations than in those with JAK2 or MPL mutations [240]. In healthy individuals, thrombopoietin (TPO) is an important cytokine/growth factor that promotes the proliferation and differentiation of megakaryocytes by binding to its cell surface receptor MPL via JAK-STAT signaling (Figure 4). In MPN patients, CRT binds and activates MPL independent of TPO, resulting in the constitutive activation of the JAK-STAT signaling pathway [241,242,243,244].

CRT has been suggested to play a role in the thrombotic process through multiple mechanisms, such as platelet activation, endothelial dysfunction, and modulation of the coagulation factors. The first reference to CRT and thrombosis was made by Kuwabara et al., who observed that wildtype CRT interacted with the endothelial layer and stimulated the release of nitric oxide, which prevented clot formation in experimentally induced coronary thrombosis in a canine model of acute arterial occlusion. Moreover, they also showed that CRT is bound to coagulation factor IX/IXa via its C-terminal region [245]. Still, it did not impact the coagulation activity of these proteins. In a subsequent study using rat models of endothelial injury in iliofemoral arteries, it was demonstrated that wildtype CRT inhibited intimal hyperplasia, reduced plaque growth, and inhibited platelet activation, potentially as a consequence of binding and interacting with factors IX and X and prothrombin. Because CRT is a Ca^2+^-binding protein, it may chelate the Ca^2+^ required by the coagulation factors for clotting activity at the sites of endothelial injury [246].

Most recently, we discovered that CRT functions as a plasminogen-binding receptor that greatly stimulates the conversion of plasminogen to plasmin [155]. Because a link between fibrinolysis defects in patients expressing CRT mutations was noted, we examined the possibility that CRT might be a plasminogen receptor that directly regulated plasmin production. Most importantly, we have observed that the two major CRT mutants, CRTdel52 and CRTins5, possessed about 40% of the plasmin-generating activity of the wildtype CRT [155]. Next, we utilized mouse embryonic fibroblasts obtained from wildtype embryos (K41) and CRT-knockout embryos (K42) to examine cellular plasmin production [129]. When we incubated K41 and K42 embryonic fibroblasts with plasminogen and measured the plasmin generation, we observed that the K42 fibroblasts displayed a drastic 90% loss in plasmin generation compared to that of the K41 fibroblasts. Our results suggest that CRT is an important cellular plasminogen regulatory protein that may contribute to the defect in clot lysis observed in ET patients. The precise mechanism of how the loss of plasmin activity impacts thrombosis in patients with CRT mutations is a focus of future investigations. However, this is the first time a protein in the plasminogen activation pathway, also known as a plasminogen receptor, has been shown to be involved in MPNs.

The CRT mutants associated with MPNs are shown to interfere with ICD and antitumor immune responses, as well as Ca^2+^-binding (calcium homeostasis) and antitumor immune responses. Several lines of evidence have suggested this. First, because of an altered C-terminal amino acid sequence and the loss of the KDEL sequence, CRT MPN mutants are secreted from cells. This results in the binding and saturation of the CRT receptors on the surfaces of phagocytic cells (DCs), which prevent them from recognizing the cell-surface CRT on dying cancer cells [247,248]. This event dampens the antitumor immune response mediated by chemotherapy or the PD-1 blockade. Those studies highlighted the potential mechanisms of cancer cells’ immune evasion and resistance to immune checkpoint inhibitor therapies in MPNs. The CRT mutants can further affect immune evasion in MPNs by decreasing the MHC class I processing and antigen presentation. They do so by compromising MHC class I’s binding to the PLC, which is crucial for MHC class I processing and antigen presentation. As described above, wildtype CRT is associated with MHC class I via a conserved glycan in the MHC I protein. This association further recruits MHC I in the PLC. Although CRT mutants continue associating with MHC I via the glycan-binding region, they do not possess the peptide loading function necessary for MHC class I processing [202]. This is potentially because of the low levels of CRT mutants in the ER, as most CRT is likely secreted. The mutants form a transient association with the PLC, but the low ER levels reduce the efficiency of the loading of MHC class I onto the complex. This ultimately leads to poor peptide-loaded MHC class I expression on the cell surface. The reduced MHC class I processing and impaired antigen presentation hinder CRT mutant MPN cells from being killed by cytotoxic T-cells. The loss of ER CRT could further limit the ability of the immune system to recognize and eliminate the MPN mutant cells, resulting in disease progression. Furthermore, there is evidence of MHC class I allele skewing in MPN patients with CRT mutants. This predominantly manifests as an underrepresentation of alleles, which can effectively present CRT mutant neoepitopes [249]. Because the C-domain of wildtype CRT is crucial for Ca^2+^ binding, disruption of this domain in the CRT MPN Type I and Type II mutants negatively impacts Ca^2+^ binding. These effects on Ca^2+^-binding are more pronounced in Type I mutations, as they eliminate all negatively charged amino acids in the C-terminus. In contrast, Type II eliminates only 50% of the negatively charged groups. Studies by Ibarra et al. have characterized the differences in the Ca^2+^-binding properties and the unfolded-protein responses (UPRs) and pathogenic outcomes between these two groups of mutations. They demonstrated that Type I CALR mutations, unlike Type II, significantly impair Ca^2+^ binding, leading to ER stress and the activation of the IRE1α/XBP1 pathway. This creates a specific vulnerability in Type I CALR-mutated MPNs that could be exploited for targeted therapy for MPNs [250].

## 8. Conclusions

This review highlights the massive contributions of many laboratories that have formed the framework of our understanding of the structure and function of CRT. A timeline of these discoveries is presented in Figure 6. This review was necessitated because most of the incremental advances in CRT research and the credit to the laboratories that contributed to those advances had been marginalized by the plethora of review articles written about CRT. Therefore, the authors of this review have painstakingly reviewed the literature and attempted to assign credit to the laboratories that have advanced the CRT field. We apologize to any researchers whose contributions have been omitted and welcome the opportunity to include those discoveries as an addendum.

We have also addressed the controversy concerning the discovery of CRT and shown that CRT was initially discovered as the Ca^2+^-binding ER protein calregulin and not as the SR protein(s) called the HACBP. The evidence clearly shows that the HACBP was likely a mixture of several SR proteins, of which more than 90% were the 53 kDa glycoprotein. Other evidence, including the lack of Stains-all staining and the distinct Ca^2+^-binding properties exhibited by the HACBP, has relegated the HACBP to the cults of antiquity.

Since the discovery of the renaming of calregulin to CRT, there has been an explosion of exciting discoveries. Within the cell, CRT has been shown to play important roles as a Ca^2+^ buffer and a chaperone. CRT also plays a number of roles in cancer biology. On the surface of cancer cells, CRT plays the role of an “eat-me” signal for the activation of phagocytosis. CRT is upregulated in many cancers and has been suggested to influence cell proliferation as well as migration and invasion. The recently discovered role of CRT as a plasminogen receptor may provide a link between CRT and invasion. CRT may be a diagnostic marker and a therapeutic target for cancer. CRT mutations associated with MPNs markedly differ in many functionalities, such as phagocytosis, ICD, and calcium binding, compared to those of the wildtype CRT. These alterations impact the development and progression of MPNs. The future holds much promise for clarifying the roles of CRT in cancer and elucidating potentially new roles.

## Figures and Tables

**Figure 1 cancers-17-00288-f001:**
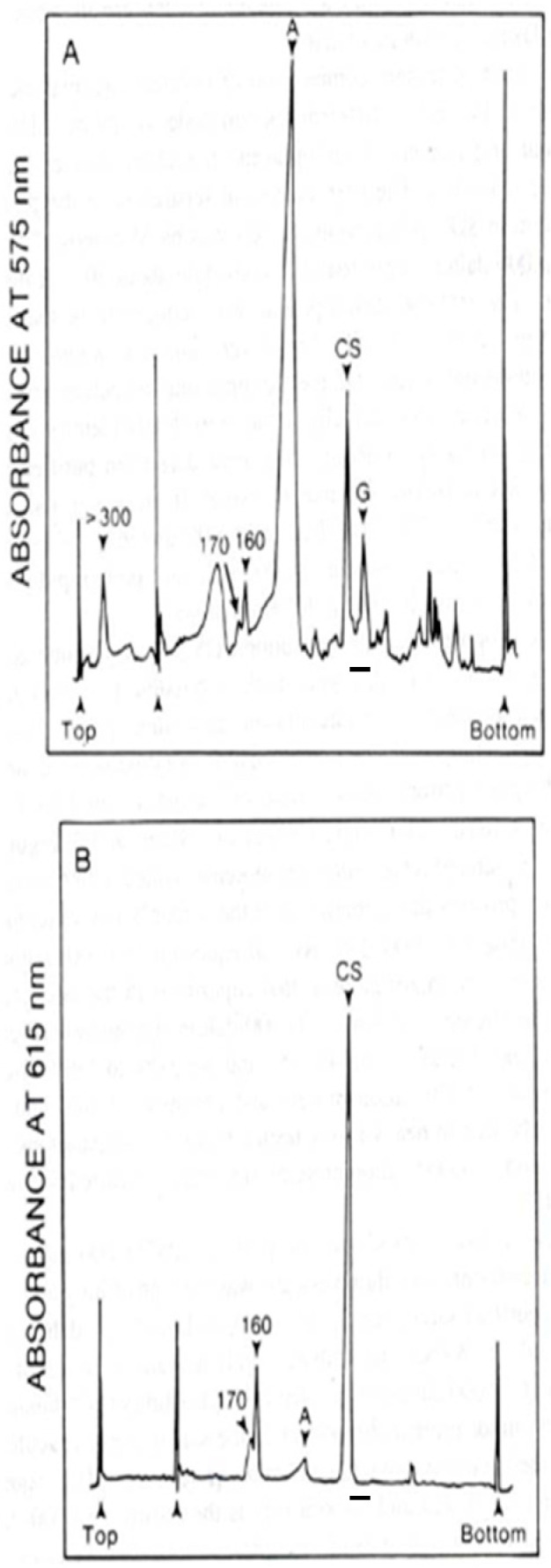
Densiometric analysis of SR proteins stained with Coomassie blue (**A**) or Stains-all (**B**) following separation of the MacLennan SR preparation by Laemmli SDS-PAGE. The highlighted proteins include the Ca^2+^ ATPase (*A*), calsequestrin (*CS*), and the 53 kDa glycoprotein (G). The figure shows intense staining of calsequestrin by Stains-all but the absence of this staining in the 55 kDa region, the region occupied by the HACBP/CRT (the region to the right of *CS*). The line indicates the positions of the bands corresponding to the HACBP. Reproduced from [5] by permission from Dr. Kevin Campbell.

**Figure 2 cancers-17-00288-f002:**
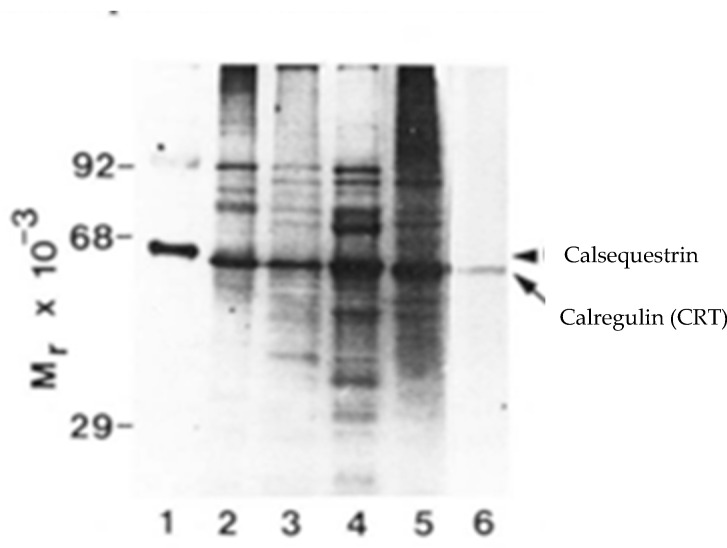
SDS-PAGE of proteins extracted from microsomes of rat skeletal muscle (1), liver (2), parotid (3), pancreas (4), and brain (5). The proteins were stained with “Stains-all” (lanes 1–5). The proteins indicated by the two arrows, calsequestrin and calregulin (calreticulin), are stained blue. Lane 6 is a western blot of lane 5, using an anti-calregulin antibody. Reprinted by permission from the Journal of Biological Chemistry [16].

**Figure 4 cancers-17-00288-f004:**
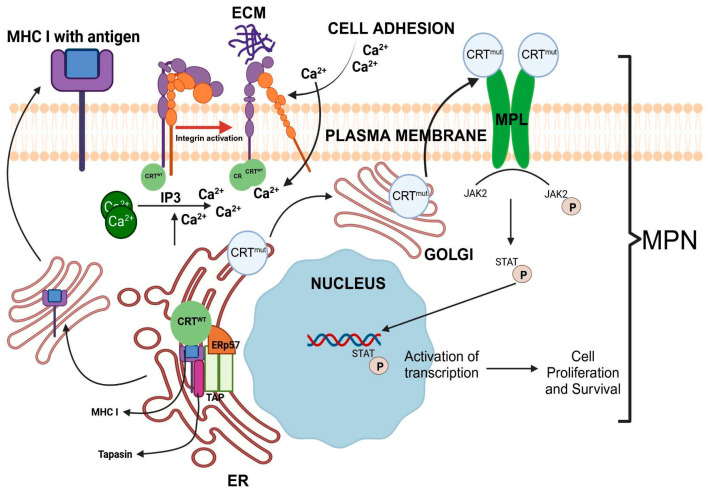
Roles of calreticulin (CRT) in multiple cellular pathways and processes. Wildtype CRT (CRTWT) plays a crucial role in major histocompatibility class I (MHC I) folding, processing, and antigen presentation. MHC 1 interacts with Erp57 in the endoplasmic reticulum (ER), along with tapasin, along with the transporter associated with antigen processing (TAP), and form the peptide-loading complex. These proteins load the antigen to MHC I and then transport the complex via the Golgi apparatus to the cell surface. CRT binds to the cytoplasmic domain of the integrin α-subunit (shown in purple), activating integrin. The activated integrin binds the extracellular matrix proteins (ECM), resulting in cellular adhesion. Upon this integrin engagement, a transient elevation in the cytosolic calcium (Ca^2+^) concentration is mediated by both Ca^2+^ influx (through a voltage-gated L-type Ca^2+^ channel—not shown) and release from intracellular Ca^2+^ stores via inositol trisphosphate (IP3). Mutant CRT (CRT^mut^), expressed by patients with myeloproliferative neoplasms (MPNs), are secreted outside the cell and bind to the thrombopoietin receptor (MPL), resulting in the activation of MPL and downstream signaling via the JAK-STAT pathway via the phosphorylation of JAK (JAK-P) and STAT (STAT-P). This results in transcriptional activation of genes involved in cell survival and proliferation, causing MPNs.

**Figure 5 cancers-17-00288-f005:**
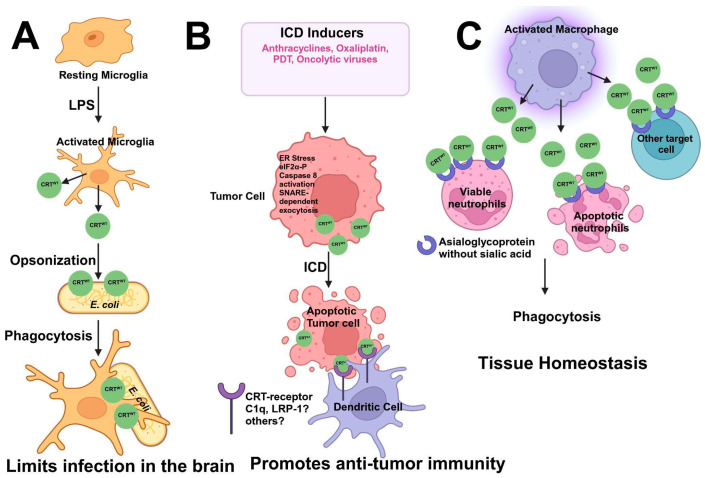
Roles of calreticulin (CRT) in phagocytosis and immunogenic cell death (ICD). (**A**) Activated microglia from the brain secrete CRT, which binds to *E. coli* (bacteria), further enhancing their phagocytosis by the microglia. This limits infection in the brain. (**B**) Immunogenic cell death (ICD) is a process that causes cancer cell death by eliciting an immune response. Several chemotherapeutic agents, such as anthracyclines and oxaliplatin, as well as photodynamic therapy (PDT) and oncolytic viruses, result in endoplasmic reticulum (ER) stress, phosphorylation of eIF-2α, activation of caspase 8, and cell-surface exposure of CRT. The cell-surface CRT is recognized by dendritic cells via receptors, such as low-density lipoprotein-receptor-related protein 1 (LRP1), C1q, and possibly other unknown proteins, which result in the engulfment of the apoptotic cancer cell. (**C**) CRT also plays a role in tissue homeostasis. Activated macrophages secrete CRT, which binds to viable or apoptotic neutrophils and other target cells via asialoglycoprotein on their cell surface, resulting in phagocytosis of these cells. This phagocytosis is absent if the asialoglycoprotein on the target cell’s surface is bound by sialic acid.

**Figure 6 cancers-17-00288-f006:**
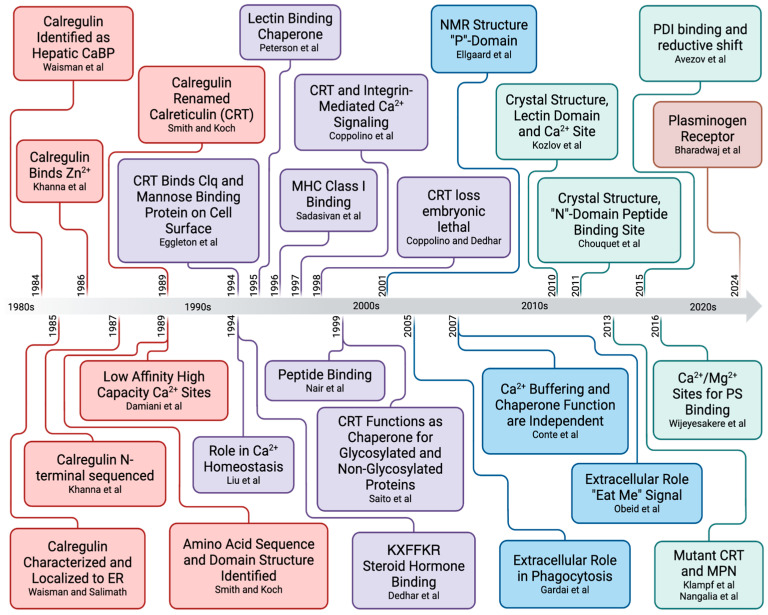
Timeline of initial discoveries in the calreticulin field. The initial reports are highlighted. The references are Waisman et al., 1984 [37]; Salimath et al., 1985 [38]; Khanna et al., 1986 [21]; Khanna et al., 1997 [39]; Damiani et al., 1989 [82]; Smith and Koch, 1989 [40]; Eggleton et al., 1994 [105]; Liu et al., 1994 [140]; Dedhar, 1994 [134]; Peterson et al., 1995 [63]; Sadasivan et al., 1996 [131]; Coppolino et al., 1997 [145]; Coppolino and Dedhar 1998 [158]; Nair et al., 1999 [64]; Saito et al., 1999 [58]; Ellgaard et al., 2001 [79]; Gardai et al., 2005 [138]; Conte et al., 2007 [86]; Obeid et al., 2007 [148]; Kozlov et al., 2010 [56]; Chouquet et al., 2011 [55]; Klampf et al., 2013 [236]; Nangalia et al., 2013 [237]; Avezov et al., 2015 [78]; Wijeyesakere et al., 2016 [87]; Bharadwaj et al., 2024 [155].

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
