# Peer review of "Calreticulin—From the Endoplasmic Reticulum to the Plasma Membrane—Adventures of a Wandering Protein"

_cancers, 2025, doi:10.3390/cancers17020288_

Round 1
Reviewer 1 Report
Comments and Suggestions for Authors
The aim of this article by Gillian Okura et al. is to recapitulate the history of calreticulin (CRT) and show its role in several cellular processes and in cancer pathology. This article has a historical flavour, insofar as a large part of the text is devoted to the discovery and early characterization of this protein back in the 70s. The result is to put the first discoverers in their rightful place. However, despite its good intentions, this review article is difficult to read, as the text is dense with very few didactic illustrations.
Major recommendations:
- The layout needs to be revised: the entire text seems to be an introduction from 1.1 to 1.9, with a short conclusion. The text needs to be reorganized and divided into more easily identifiable sections.
- Even if the first two figures are historically relevant, the text should be illustrated with other graphic elements. Indeed, part 1.2 on protein structure would certainly be easier to follow with an illustration of the protein's three-dimensional structure (PDB viewer).
- Draw a diagram of the signaling functions of CRT.
As a final remark, the magazine “Cancers” deals with pathology. How can we justify the publication of such a magazine that talks so little about it? I reorder to develop this part, even if it means reducing the space dedicated to the initial characterization.
Author Response
Reviewer 1
Thank you for your comments and your detailed review. We have complied with all your requests where possible and agree that these additions have strengthened the manuscript.
The aim of this article by Gillian Okura et al. is to recapitulate the history of calreticulin (CRT) and show its role in several cellular processes and in cancer pathology. This article has a historical flavour, insofar as a large part of the text is devoted to the discovery and early characterization of this protein back in the 70s. The result is to put the first discoverers in their rightful place. However, despite its good intentions, this review article is difficult to read, as the text is dense with very few didactic illustrations.
Major recommendations:
- The layout needs to be revised: the entire text seems to be an introduction from 1.1 to 1.9, with a short conclusion. The text needs to be reorganized and divided into more easily identifiable sections.
We have added several section breaks as requested. They are: 1.2 Does the HACBP actually exist?, 1.3 Are the HACBP and CRT the same protein?, 1.4 Calreticulin-Gene structure, 1.5 CRT domain structure, 1.6 CRT metal ion-binding properties. This has divided the long text into shorter sections.
- Even if the first two figures are historically relevant, the text should be illustrated with other graphic elements. Indeed, part 1.2 on protein structure would certainly be easier to follow with an illustration of the protein's three-dimensional structure (PDB viewer).
We have included this figure as Figure 3. As requested by another reviewer, we have also included a Table listing the interacting partners and a separate table listing the function of CRT. This will alleviate the density of text issues.
- Draw a diagram of the signaling functions of CRT.
We have included this figure as Figure 4, 5.
As a final remark, the magazine “Cancers” deals with pathology. How can we justify the publication of such a magazine that talks so little about it? I reorder to develop this part, even if it means reducing the space dedicated to the initial characterization.
--In contrast to myeloproliferative neoplasms, driver mutations in CALR have not been associated with other hematological or solid tumors (as yet). When those associations become apparent we are happy to include them in a subsequent review or an addenda to this review. I did not feel that a discussion of the more than 100 single nucleotide polymorphism sites that are known to be distributed along the CALR exons, that have been described as somatic mutations of the CALR gene, were within the scope of the review, unless a direct link to cancer could be discussed.
Reviewer 2 Report
Comments and Suggestions for Authors
The authors succeeded in presenting of the CRT research history in the form of a rather exciting, almost detective story, and also provided with significant modern information about its properties and biological functions. Of course, the systematization and generalization of extensive, sometimes contradictory information about the structure and functions of this protein in the form of a review is of a certain scientific value.
However, a number of issues remain that need to be clarified in order to improve the quality of the manuscript
- As for the historical retrospective, in my opinion, this way of presenting information is quite interesting, especially for young researchers, who find it very useful to learn about the previously used methodologies and the possibility of interpreting the results obtained. It would be very good to emphasize the accuracy of the methods used in the primary experiments, as well as how much more informative modern technologies (such as mass spectrometry, genomic technologies, etc.) are.
- fig 1, it is not clear what was used as reference proteins for comparison, perhaps the original source contains this data.
- Taking into account the accuracy of SDS-PAGE of about 10%, how can the authors explain that proteins with apparent Mol masses of 46 and 55 KDa were not separated.
- it will be convenient for the reader if the authors add the Kd value in addition to the range, line 257 and so on in the text.
-“hairpin-like structure” as well as « ORAI» should be written uniformly throughout the manuscript.
- Based on the context, the sentence «The P-domain also interacts with 294 other proteins that function as ER chaperones and form ER chaperone-substrate complexes such as ERp29 [71], CypB [72], PDI1A [73–75], and ERp57 [60,76,77]” unnecessarily contains “also”.
- Poorly «the activation state of integrin»
- Reference 5 should be drawn up according to the requirements of the journal.
More Significant Issues:
- Since the period of study that the authors are considering is very long, it is fundamentally important to indicate the method by which the given characteristics or functional features of the protein were obtained. In addition, specifying the method by which the data were obtained indicates to the reader what data exactly and with what accuracy/reliability it can provide. In some cases, the authors share this information, although in many cases they only provide interpretations.
- There is confusion in the terminology use that interferes with the perception of information
For example, the term DOMAIN is used in its generally accepted structural sense (lines 247-252) then followed by "polypeptide binding domain" (line 270). Authors should clarify what they mean by the term DOMAIN and follow this throughout the manuscript.
It should be clarified that the authors mean «This region, along with the central N-domain of the protein» (lines 296-297).
-The authors probably do not know the established meaning of the term LIGAND when describing the interactions of substances with proteins, which in this case does not consider the amino acid residues of proteins that form the binding sites as LIGANDS.

Author Response
Thank you for your comments and your detailed review. We have complied with all your requests where possible and agree that these additions have strengthened the manuscript.
The authors succeeded in presenting the CRT research history in the form of a rather exciting, almost detective story and also provided significant modern information about its properties and biological functions. Of course, the systematization and generalization of extensive, sometimes contradictory information about the structure and functions of this protein in the form of a review is of a certain scientific value.
However, a number of issues remain that need to be clarified in order to improve the quality of the manuscript
- As for the historical retrospective, in my opinion, this way of presenting information is quite interesting, especially for young researchers, who find it very useful to learn about the previously used methodologies and the possibility of interpreting the results obtained. It would be very good to emphasize the accuracy of the methods used in the primary experiments, as well as how much more informative modern technologies (such as mass spectrometry, genomic technologies, etc.) are.
Because of the general readership of the review, we were afraid of boring the audience with too much detail about methodologies but were careful to include the key references. As a die-hard biochemist, I totally agree with your comments on the importance of understanding the accuracy and limitations of the techniques used. I would respectfully submit that much of the precision of these techniques is indicated by the use of statistics, which, in most early studies, is missing. As for determining the accuracy of the various techniques, I respectfully submit that it is out of the scope of the review. For example, we avoided a discussion of the accuracy of equilibrium dialysis with other techniques, such as flash photolysis of caged Ca2+ or stopped-flow and 43Ca2+-NMR or isothermal titration calorimetry. However, we have addressed your request by clearly defining the techniques used. To accommodate your request, we have added to the text; Line 138-The basic difference between these two methods is that the Weber and Osborne method is a continuous electrophoresis method in which proteins are electrophoresed in a single buffer. In contrast, the Laemmli system is a discontinuous system that utilizes a stacking gel on top of a resolving gel. The stacking gel generates a tight voltage gradient between the leading edge of one buffer and the trailing edge of a second buffer, resulting in proteins of different charges forming narrow bands or discs, driven by the voltage gradient at the discontinuity. As such the Laemmli procedure results in tighter protein bands. Line 216-The studies of Ca2+-binding by CRT were performed using the equilibrium dialysis method ).
- fig 1, it is not clear what was used as reference proteins for comparison, perhaps the original source contains this data.
Unfortunately, the reference proteins were not provided in the paper.
- Taking into account the accuracy of SDS-PAGE of about 10%, how can the authors explain that proteins with apparent Mol masses of 46 and 55 KDa were not separated.
Great question. I wondered why the Weber and Osborn SDS PAGE method, as opposed to the Laemmli methods, failed to separate these 55 kDa proteins since the SDS should load the SR proteins similarly in both systems. My suspicion is that the stacking gel of the Laemmli system results in a better resolution of proteins before they enter the running gel. The use of a gradient running gel also improves resolution. That, however does not explain the MW shift of calsequestrin in the two systems. However, I think that the review does point out that Dr. MacLennan made the mistake of assuming that a single band on SDS PAGE corresponds to a single protein–this should be useful to young investigators as a lot of the fundamentals about techniques are not taught in enough detail in courses (in my opinion).
- it will be convenient for the reader if the authors add the Kd value in addition to the range, line 257 and so on in the text.
Absolutely–we have added Table II to provide these details.
-“hairpin-like structure” as well as « ORAI» should be written uniformly throughout the manuscript.
Corrected
- Based on the context, the sentence «The P-domain also interacts with 294 other proteins that function as ER chaperones and form ER chaperone-substrate complexes such as ERp29 [71], CypB [72], PDI1A [73–75], and ERp57 [60,76,77]” unnecessarily contains “also”.
Corrected
- Poorly «the activation state of integrin»
Changed to “have impaired integrin-mediated adhesion to ECM. “
- Reference 5 should be drawn up according to the requirements of the journal.
Corrected
More Significant Issues:
- Since the period of study that the authors are considering is very long, it is fundamentally important to indicate the method by which the given characteristics or functional features of the protein were obtained. In addition, specifying the method by which the data were obtained indicates to the reader what data exactly and with what accuracy/reliability it can provide. In some cases, the authors share this information, although in many cases they only provide interpretations.
As also requested by a reviewer–Because of the general readership of the review, we were afraid of boring the audience with too much detail about methodologies but were careful to include the key references. As a die-hard biochemist, I totally agree with your comments on the importance of understanding the accuracy and limitations of the techniques used. I would respectfully submit that much of the precision of these techniques is indicated by the use of statistics, which, in most early studies, is missing. As for determining the accuracy of the various techniques, I respectfully submit that it is out of the scope of the review. For example, we avoided a discussion of the accuracy of equilibrium dialysis with other techniques, such as flash photolysis of caged Ca2+ or stopped-flow and 43Ca2+-NMR or isothermal titration calorimetry. However, we have addressed your request by clearly defining the techniques used. To accommodate your request, we have added to the text; Line 138-The basic difference between these two methods is that the Weber and Osborne method is a continuous electrophoresis method in which proteins are electrophoresed in a single buffer. In contrast, the Laemmli system is a discontinuous system that utilizes a stacking gel on top of a resolving gel. The stacking gel generates a tight voltage gradient between the leading edge of one buffer and the trailing edge of a second buffer, resulting in proteins of different charges forming narrow bands or discs, driven by the voltage gradient at the discontinuity. As such the Laemmli procedure results in tighter protein bands.
Line 216-The studies of Ca2+-binding by CRT were performed using the equilibrium dialysis method.
I wondered why the Weber and Osborn SDS PAGE method, as opposed to the Laemmli methods, failed to separate these 55 kDa proteins since the SDS should load the SR proteins similarly in both systems. My suspicion is that the stacking gel of the Laemmli system results in a better resolution of proteins before they enter the running gel. The use of a gradient running gel also improves resolution. That, however does not explain the MW shift of calsequestrin in the two systems. However, I think that the review does point out that Dr. MacLennan made the mistake of assuming that a single band on SDS PAGE corresponds to a single protein–this should be useful to young investigators as a lot of the fundamentals about techniques are not taught in enough detail in courses (in my opinion).
- There is confusion in the terminology use that interferes with the perception of information
For example, the term DOMAIN is used in its generally accepted structural sense (lines 247-252) then followed by "polypeptide binding domain" (line 270). Authors should clarify what they mean by the term DOMAIN and follow this throughout the manuscript.
It should be clarified that the authors mean «This region, along with the central N-domain of the protein» (lines 296-297).
We use the term domain to denote a structural or functional region of the protein. We use it for a structural region. I agree that this term has been used less rigorously in the literature than it should be. –I have changed it to “region or site” where necessary–eg. “polypeptide binding domain” to “site”–thank you for catching these errors.
-The authors probably do not know the established meaning of the term LIGAND when describing the interactions of substances with proteins, which in this case does not consider the amino acid residues of proteins that form the binding sites as LIGANDS.
Agreed-This has been corrected –the amino acids contributing to the binding of Ca2+ have been changed to–”This Ca2+ is coordinated by seven oxygen atoms contributed by the bidentate side chain of Asp328, backbone carbonyls of Gln26, Lys62, and Lys64, and two water molecules”. It has been corrected in two other places–line 427, and 435. –I meant to state that the atoms of the amino acids provide ligands. Thank you for catching that.
Reviewer 3 Report
Comments and Suggestions for Authors
The review is devoted to calreticulin (CRT), kDa Ca2+-binding protein of the endoplasmic reticulum (ER). In the introduction, the authors describe the rather long history of the discovery of this protein. They even present a graphical timeline of initial discoveries in the calreticulin field. The authors then discuss the physicochemical properties of CRT, including its Ca2+ and Zn2+ binding properties. It seems to me that it would be useful in these chapters to provide a drawing of the three-dimensional structure of CRT, highlighting its three domains and indicating the location of the metal ion binding sites. The authors have given much attention to the possible functions of CRT inside and outside cells. Since CRT is a multifunctional protein, it would be convenient for the reader to see a listing of these functions in a single table. In addition, it would be useful to compile a table of proteins with which CRT can interact, indicating the known binding constants. The authors discussed in some detail the involvement of CRT in various diseases, including cancer. The review contains 214 literature references (27 sources within the last 7 years). I believe that the presented review will be useful to a wide range of readers of various specialties.
Author Response
Thank you for your comments and your detailed review. We have complied with all your requests where possible and agree that these additions have strengthened the manuscript.
The review is devoted to calreticulin (CRT), kDa Ca2+-binding protein of the endoplasmic reticulum (ER). In the introduction, the authors describe the rather long history of the discovery of this protein. They even present a graphical timeline of initial discoveries in the calreticulin field. The authors then discuss the physicochemical properties of CRT, including its Ca2+ and Zn2+ binding properties. It seems to me that it would be useful in these chapters to provide a drawing of the three-dimensional structure of CRT, highlighting its three domains and indicating the location of the metal ion binding sites.
We have included this figure as fig 3.
The authors have given much attention to the possible functions of CRT inside and outside cells. Since CRT is a multifunctional protein, it would be convenient for the reader to see a listing of these functions in a single table.
This has been included as Table III.
In addition, it would be useful to compile a table of proteins with which CRT can interact, indicating the known binding constants.
Table II has been added which shows the major binding partners for CRT and where possible the Kd of interaction.
The authors discussed in some detail the involvement of CRT in various diseases, including cancer. The review contains 214 literature references (27 sources within the last 7 years). I believe that the presented review will be useful to a wide range of readers of various specialties.
Thank you for these positive comments.
Round 2
Reviewer 1 Report
Comments and Suggestions for Authors
Here I have some concerns about the proposed answers to my requests. This review article does not fit with your editorial objective.
I had asked for a clear plan, it hasn't been changed, if we stick to the text it's an introduction (paragraph 1) with 13 parts (1.1 to 1-13) and a conclusion. I asked for modifications, and ... nothing!
I asked for a development of the pathological aspect associated with cancers, and the answer is bizarre, explaining that CARL mutations are not associated with cancers. In fact, why did they specifically choose this review? « -In contrast to myeloproliferative neoplasms, driver mutations in CALR have not been associated with other hematological or solid tumors (as yet). When those associations become apparent we are happy to include them in a subsequent review or an addenda to this review. I did not feel that a discussion of the more than 100 single nucleotide polymorphism sites that are known to be distributed along the CALR exons, that have been described as somatic mutations of the CALR gene, were within the scope of the review, unless a direct link to cancer could be discussed. »
Author Response
I understand the comments of the corresponding author. However, some of my remarks do not seem to have been understood. The plan: if the introduction is 1, and the following parts are 1.1, 1.2, 1.3 ... am I to understand that all the text between 1.1 and 1.13 are only introductory parts? I don't think the authors have understood my request. I just want them to change this point: 1= introduction and 2= ... I'm well aware that this article was greatly improved after the revision. However, I wasn't sure whether this article is suitable for this journal, and asked the editor to decide.Sorry for the confusion--we have renumbered the manuscript as per your request. The issue of the the link to cancer has been addressed in detail in our response that was requested by the academic editor.
.
Reviewer 2 Report
Comments and Suggestions for Authors
The authors have significantly revised the manuscript and made the necessary changes, which has led to a significant improvement in its quality.
Author Response
The authors have significantly revised the manuscript and made the necessary changes, which has led to a significant improvement in its quality.
Thank you for your positive comments and helpful suggestions.
Round 3
Reviewer 1 Report
Comments and Suggestions for Authors
This article is suitable for publication.